# Preparation and Magnetic Properties of CoFe_2_O_4_ Oriented Fiber Arrays by Electrospinning

**DOI:** 10.3390/ma13173860

**Published:** 2020-09-01

**Authors:** Chen Cheng, Jianfeng Dai, Zengpeng Li, Wei Feng

**Affiliations:** 1School of Science, Lanzhou University of Technology, Lanzhou 730050, China; chengchen2021@163.com (C.C.); fengwei8576@163.com (W.F.); 2State Key Laboratory of Advanced Processing and Recycling of Nonferrous Metals, Lanzhou University of Technology, Lanzhou 730050, China; sicnulzp@163.com; 3Key Laboratory of Solar Power System Engineering, Vocational and Technical College Jiuquan, Jiuquan 735000, China

**Keywords:** electrospinning, aligned nanofibers, nanoparticle, anisotropy, micromagnetic simulation

## Abstract

The morphology of magnetic materials has a great influence on the properties, which is attributed to the magnetic anisotropy of the materials. Therefore, it is worth studying the fabrication of the aligned fiber and the change of its domain distribution. Nanoparticles and nanofibers were prepared by the hydrothermal and electrospinning methods, respectively. At the same time, the arranged nanofibers were collected by the drum collecting device. After the same annealing at 700 °C, it was found that the diameter of fibers collected by different collecting drums is similar. By studying the hysteresis loops of nanoarrays, it was found that they had strong anisotropy. The easy axis was parallel to the long axis, the Hc and Mr of the easy axis and the hard axis were 1330.5 Oe, 32.39 Am^2^/kg, and 857.2 Oe, 24.8 Am^2^/kg, respectively. Due to the anisotropy of the shape and the interaction between the particles, the Hc could not be enhanced. Therefore, the Ms and Hc of the nanoparticles were 80.23 Am^2^/kg and 979.3 Oe, respectively. The hysteresis loop and the change of magnetic moment during the demagnetization of the CoFe_2_O_4_ nanofiber array were simulated via micromagnetic software. The simulated Hc was 1480 Oe, which was similar to the experimental value.

## 1. Introduction

With the development of science and technology, ferrite materials are known as promising materials. Due of their good magnetism, different applications in electrical and optoelectronic devices, significant resistance, and low eddy current loss. Cobalt ferrites are known as attractive magnetic materials in other oxides due to their special properties and low-cost production. Cobalt ferrite has good mechanical hardness, high coercive force, wear anisotropy, high mechanical strength, medium saturation magnetization, high chemical stability, and high-temperature magnetic order [1]. Its application in electric and photoelectric devices and high-density magnetic materials has been given increasing attention. Furthermore, it has broad application prospects in drug delivery, hyperthermia, magnetic resonance imaging, magnetic sensitivity, and tissue imaging [2].

In recent years, the applications of magnetic nanomaterials in ultra-high-density magnetic recording media [2,3], biosensors [4], magnetoelectric materials [5,6] micromagnetic devices, and microwave absorption applications [7,8,9,10] have been extensively studied. Therefore, the preparation of aligned nanofibers has important engineering application value in the sensor, electronic, and tissue engineering fields. Aligning nanofiber components will play a vital role in future nanotechnology development [11,12]. The magnetic domain arrangement of nanoparticles, disordered, and aligned nanofibers are the focus of current research. In the conventional methods of preparing aligned nanomaterials, photolithography [13], electrodeposition [14,15], and linear template [16] are usually used. Others have prepared aligned nanofibers by using an external magnetic field [17].

However, these methods require complex equipment and expensive instruments for precise preparation, and the length of samples prepared is limited. Electrospinning technology [18,19] has the advantages of simple preparation equipment, low spinning cost, and controllable process, and has become one of the main ways to effectively prepare nanofibers. By changing the receiving device of electrospinning, controlling the electric field, and including an additional magnetic field, we can obtain different forms of aligned nanofiber assemblies including oriented nanofiber membranes, oriented nanofiber bundles, and oriented nanofiber coated yarns, which will further expand the application field of directional nanofiber [20,21,22,23,24,25].

Due to the uncontrollable spray of the fibers, it is still challenging to prepare nanofiber arrays. In the electrospinning process, the filamentation process of polymer nanomaterials includes three stages: (1) the beginning and extension of the charged jet along a straight line; (2) the increase of electric bending instability and the further extension of jet; and (3) the jet solidifies into one-dimensional polymer nanomaterials on the collecting drum [26].

By adjusting the collection device of the electrospinning method, directional fibers can be obtained. The parallel electrode collecting device is a simple method to prepare oriented fibers. Park et al. [27] placed the aluminum sheet obliquely, which provided sufficient time and space for the self-stretching of nanofibers and improved the orientation of nanofibers. Liu et al. [28] used a similar device, but they introduced an external magnetic field in the collector area to achieve better electrospinning of ordered nanofibers. In order to make the device have a better collection effect, Yang et al. [29] also added magnetic materials in the body before the experiment. However, the collection area of this method is narrow, which cannot realize the preparation of large-area highly oriented nanofiber membranes, and there are still limitations in thickness and length. Li et al. [30] prepared multilayer multidirectional nanofiber films by multi-electrode collection methods. This further broadens the scope of morphology and application of oriented nanofibers. The disadvantage is that the collection range of the nanofiber membrane is narrow and the nanofiber membrane is small. Stretched copper wire between two disk edges as a collection device was performed by Katta et al. [31]. As the cylinder rotates slowly, the next copper wire attracts the nanofibers, which stretch the copper wires vertically across the gap between wires. The device is simple and can prepare oriented nanofiber bundles, but the collection range and thickness are limited.

Through the above analysis, in order to make the device simple and the collection effect better, we will use the drum collecting device. We believe that the Taylor cone angle can be reduced by reducing the collection width of the collection bucket to make the spray nanofibers more concentrated, or by changing the linear speed of the drum, so the rotation speed and the filament winding speed are close to each other. The Taylor taper refers to the fact that the solution is subjected to surface tension and the electric field force at the nozzle of the needle tube. When the voltage reaches the critical value, the solution at the needle nozzle will become conical [32].

In this paper, CoFe_2_O_4_ nanofibers with a high orientation will be prepared by controlling the diameter and width of the drum collecting device. The influence of the shape parameters of the collecting drum on the fiber shape orientation was further discussed and verified. The particles’ diameters, which were similar to the fibers’ diameters, were prepared by the hydrothermal method. The magnetic properties of nanofibers and nanoparticles were compared. So that the pollution and costs are reduced, our solvent only uses water and alcohol to achieve environmental friendliness. Finally, the domain changes of the CoFe_2_O_4_ nanofiber array during demagnetization were simulated by the object oriented micro magnetic framework (OOMMF).

## 2. Experimental Details

All other chemicals used in this work were of analytical grade. The Co(NO_3_)_3_·6H_2_O (Aladdin, Shanghai, China) and Fe(NO_3_)_3_·9H_2_O (Aladdin, Shanghai, China) with a Co/Fe molar ratio of 1:2 were dissolved in 5 mL deionized water and ethanol (Alfa-Aesar, Haverhill, MA, USA). Then, 0.5 g polyvinyl pyrrolidone (PVP K_90_, M_w_ = 1,300,000, Aladdin, Shanghai, China) was added to the precursor solution and stirred for 6 h. A syringe pump was used to deliver the precursor solution to a stainless steel needle with a constant flow rate of 0.3 mL/h. The needle was connected to a high-voltage power supply. In our experiment, the voltage and the distance between the syringe needle and the grounding collector were 17 kV and 15 cm, respectively. The speed of the collecting drum was 1500 r/min. Finally, the precursor nanofibers were cut into small pieces and placed in a tubular furnace at 700 °C for 4 h heat treatment. The treatment was carried out in an air atmosphere. The experimental principle of electrospinning is shown in Figure 1a. The colored spheres in Figure 1 represent different components. Under the constraint of the polymer, nanofibers were made. In this experiment, three collecting drums were used: (1) the collecting width was 4 cm and the diameter of the collecting drum was 7 cm; (2) the collecting width was 2 cm and the diameter of the collecting cylinder was 7 cm; and (3) the collecting width was 4 cm and the diameter of the collecting cylinder was 14 cm. The samples were named CFO-F-W4-D7, CFO-F-W2-D7, and CFO-F-W4-D14, respectively.

Hydrothermal synthesis of nanoparticles is a process in which Co(NO_3_)_3_·6H_2_O (Aladdin, Shanghai, China) and Fe(NO_3_)_3_·9H_2_O (Aladdin, Shanghai, China) with a Co/Fe molar ratio of 1:2 were added to 40 mL deionized water and dissolved completely, then the solution was poured into the reactor. The reactor was held at 200 °C for 4 h and then sintered at 700 °C for 4 h. The treatment was carried out in an air atmosphere. A sample of nanoparticles was obtained and were named CFO-P.

The phase composition of the samples was analyzed by a Rigaku d/max-2400 rotating X-ray diffractometer (XRD, Rigaku, Tokyo, Japan). The morphology and energy dispersive spectrometer (EDS) pictures of the samples were observed by a JEM-6701F scanning electron microscopy (SEM, JEOL, Tokyo, Japan). The magnetic properties were measured by a MicroSense EV-9 vibrating sample magnetometer (VSM, MicroSense, Lowell, MA, USA).

## 3. Results and Discussion

### 3.1. XRD Analysis

The structural characteristics of three kinds of CoFe_2_O_4_ nanofibers and nanoparticle samples calcinated at 700 °C were analyzed by XRD. The XRD spectra of these four samples are given as shown in Figure 1b. All peaks were of the spinel phase index, using standard (JCPDS) card number PDF 22-1086. The results showed that the XRD peaks of the four samples were consistent with the standard peaks, and no other peaks were found. The results show that the sample was pure spinel material.

### 3.2. Morphological Analysis

Figure 2a–d showed the SEM characterization of three kinds of nanofibers and nanoparticles annealed at 700 °C. The illustrations are the high magnification images of the selected area. Through Figure 2, it was found that the average diameters of three nanofibers were 94.6, 96.6, and 99.8 nm, respectively, with no other morphology and long-range and smooth characteristics. From Figure 2a, it can be concluded that CFO-F-W4-D7 does not produce aligned nanofibers, and the fibers are in disorder. As can be seen from Figure 2b, CFO-F-W2-D7 had a higher degree of orientation than CFO-F-W4-D7. This is because reducing the collection width of the collection tube will make the electric field distribution become centralized so that the nanofibers will be more centralized and the directionality of the nanofibers will be improved. Figure 2d shows that CFO-P nanoparticles are spherical and have no fixed orientation. The particle size was about 90 nm.

From Figure 2c, we can see that CFO-F-W4-D14 nanofibers had a high orientation. Although they were not fully aligned, CFO-F-W4-D14 had a higher degree of orientation than CFO-F-W4-D7 or CFO-F-W2-D7. This shows that in the preparation process, with the same rotational speed, increasing the linear velocity of the drum can reduce the spray range and the influence of fiber vibration on the arrangement of nanofibers, so as to obtain a high arrangement of nanofibers. Figure 1c shows the SEM image of the CFO-F-W4-D14 sample at low magnification. CFO-F-W4-D14 has high directivity and can reach centimeters in length. EDS analysis shows that iron, cobalt, and oxygen elements were contained in CFO-F-W4-D14, which is consistent with the XRD analysis.

### 3.3. Magnetic Performance Analysis

For nanofibers, alignment fibers and disordered fibers have great changes in magnetic properties due to the influence of shape anisotropy. Figure 3 provides the orthogonal axis hysteresis loops of non-oriented and aligned nanofibers at room temperature. All nanofibers exhibited typical smooth single-phase hysteresis loops. The bottom illustrations in Figure 3a,b show the SEM images of non-directional and aligned nanofibers, with arrows indicating parallel and perpendicular field configurations.

Table 1 shows the magnetic properties of non-oriented and aligned nanofibers. The hysteresis loops of non-oriented nanofibers parallel and perpendicular to the axis have similar coercivity (Hc), remanence (Mr), saturation magnetization (Ms), and rectangular ratio (Mr/Ms). Hc was 1126.2 Oe and 1118.2 Oe, and Mr was 24.83 Am^2^/kg and 25.12 Am^2^/kg. For aligned nanofibers, the Hc of the parallel field configuration and perpendicular field configuration was 1330.5 Oe and 857.2 Oe, respectively, while the Mr was 32.39 Am^2^/kg and 24.80 Am^2^/kg, respectively. In addition, the Mr/Ms of the parallel arrangement was higher than that of the perpendicular arrangement. This shows that the aligned nanofibers had obvious magnetic anisotropy.

In order to observe the difference between anisotropy and isotropy more clearly and intuitively, the M_r_ changes of the two samples at 0–180° (5° apart from each point) were measured. The trend of M_r_ in the 180° field was obtained by fitting the points. Figure 3c shows that for the 180° M_r_ of the non-oriented nanofibers, it was found that the curve fitted by each point is straight, indicating that non-oriented nanofibers have isotropy. Figure 3d shows that the 180° M_r_ curve of aligned fibers was a parabola, indicating that the aligned nanofibers have anisotropy. This means that aligning of the hard and easy-to-magnetize axes of nanofibers were perpendicular and parallel to the long axes, respectively. Although the anisotropy of nano-magnetic materials is the result of the interaction between shape anisotropy and magnetocrystalline anisotropy, shape anisotropy has a greater influence on magnetic anisotropy for aligned nanofibers.

For magnetic nanoparticles and nanofibers, their magnetic properties are different due to the interaction of shape anisotropy. The hysteresis loops and magnetic properties of the CoFe_2_O_4_ nanoparticles and nanofibers prepared by the hydrothermal method and electrospinning method are shown in Figure 4 and Table 1, respectively. Found from M_s_ data, when magnetic nanoparticles are magnetized, there are no other interaction constraints, making it easy to be magnetized by the external magnetic field, so it has a high saturation magnetization. However, for nanofibers, due to the different magnetic domain arrangements from the nanofibers, in the magnetization process, nanofibers interact with adjacent nanofibers to minimize energy, in addition to the long-axis particles. As a result, the M_s_ of the nanofibers was smaller than that of the nanoparticles. It is for these reasons that nanofibers have a higher M_r_/M_s_ ratio.

We found that the nanofibers had higher H_c_ than the nanoparticles. This is due to the fact that under the influence of shape anisotropy, the domains of nanofibers are connected in series along the long axis. When the material is in the demagnetization state, on account of the arrangement and distribution of magnetic domains in the nanofibers, the magnetic induction strength of nanofibers is affected by the morphology of nanofibers, so it is difficult to reduce the magnetic induction strength of nanofibers to zero. Therefore, the nanoparticles are easier to demagnetize and have lower H_c_.

Similar findings were found in the studies by Li [33] and Mordinas [34]. Table 2 shows the magnetic properties of CoFe_2_O_4_ with different morphologies. It was found that the fiber had high H_c_. This was due to the longest axial ratio of one-dimensional nanomaterials, so the shape anisotropy of fibers must be considered. It was found that the system with a magnetic dipole in the linear chain will enhance the coercivity. This dipole–dipole interaction between grains in one-dimensional fibers plays an important role in the magnetization process [33]. Therefore, a higher magnetic field is needed to overcome this anisotropy. As above-mentioned, increasing the orientation of nanofibers has a great influence on the magnetic properties. This will further expand the application of CoFe_2_O_4_ nanofibers in new fields such as biomagnetism, magnetic recording materials, and microwave absorbers, and is of great significance [35,36,37,38].

## 4. Discussion

The magnetic properties and magnetization reversal process of CoFe_2_O_4_ nanofiber array were studied using OOMMF micromagnetic simulation technology. CoFe_2_O_4_ exhibits hard magnetic behavior with positive and a much larger magnetocrystalline anisotropy constant, which leads to <100> as the easy-axis and <111> as the hard-axis [45,46]. The dependence of magnetization (M) on the applied magnetic field (H) can be expressed as Equation (1) [47]:(1)$${M(H)=M}_{s}(1-\frac{0{.07619K}^{2}}{H^{2}M_{s}^{2}}-\frac{0{.0384K}^{3}}{H^{3}M_{s}^{3}})$$
where M_s_ is the saturation magnetization and K is the effective magnetic anisotropy. The first numerical coefficient value of 0.07619 was used due to the cubic anisotropy of CoFe_2_O_4_. Using the experimental data of aligned nanofibers with the field parallel to the fiber direction, M_s_ = 2.65 × 10^5^ A/m (ideal cobalt ferrite density was 5.3 g/cm^3^, from JCPDS card number PDF 22-1086), H_c_ = 0.13 mT, the effective magnetic anisotropy value K of CoFe_2_O_4_ nanofibers was about 0.836 × 10^5^ J/m^3^. In order to verify the experimental process, the simulated external magnetic field took 0.1 kOe as the step and scanned the magnetic field from +20 kOe to −20 kOe. When the magnetization was rapidly relaxed to equilibrium, the damping constant α was set to 0.5. On the basis of the finite difference method, the size of the fiber cell should be lower than the exchange length ($l_{\mathrm{exch}}=\sqrt{2{A/\mu }_{0}M_{s}}$), and a single nanowire was divided into several 3 nm × 3 nm × 3 nm units. In this simulation, the diameter of the CoFe_2_O_4_ nanofiber model was set to 99 nm and the maximum length was 999 nm. The results were obtained within the appropriate calculation time [48]. Figure 5a shows the hysteresis loop of the CoFe_2_O_4_ nanofibers obtained by simulation and experiment. The H_c_ of the model was 1480 Oe and the experimental value was 1330.5 Oe.

To further understand the magnetization reversal mechanism of the nanofibers, the demagnetization process of the CoFe_2_O_4_ nanofiber array was studied. Figure 5b shows a vertical view of the magnetic moment distribution of the nanofiber array in different states. From the vertical view of the array, in the simulation process, it was found that the magnetic moment of some nanofibers in the array was reversed in an instant, while other nanofibers need a higher external magnetic field. Due to the stray field of neighboring cylindrical nanofibers, an external field was added, which led to a higher field and made it easier to magnetize. After that, these inverted nanofibers produced a stray field that was the opposite to the external field, so a higher external magnetic field was needed to change the magnetic moment [49,50]. This conversion mechanism is consistent with the demagnetization state of the Co nanofiber array described by Li Hongjian [51].

## 5. Conclusions

By comparing the magnetic properties of non-oriented nanofibers with oriented nanofibers, it was found that aligned nanofibers are anisotropic. Due to shape anisotropy, the hard axis and easy axis of the magnetization axis were perpendicular and parallel to the long axis. By comparing the magnetic properties of nanofibers with nanoparticles, it was found that the difference between them was mainly due to the different morphology. The hysteresis loop and domain motion in the demagnetization process of nanofibers were simulated by the micro magnetic software, and a similar H_c_ was obtained. In the simulation process, it was found that the magnetic moment of some nanofibers in the array was reversed in an instant, while the other nanofibers needed a higher external magnetic field, and the demagnetization process was finally completed.

## Figures and Tables

**Figure 1 materials-13-03860-f001:**
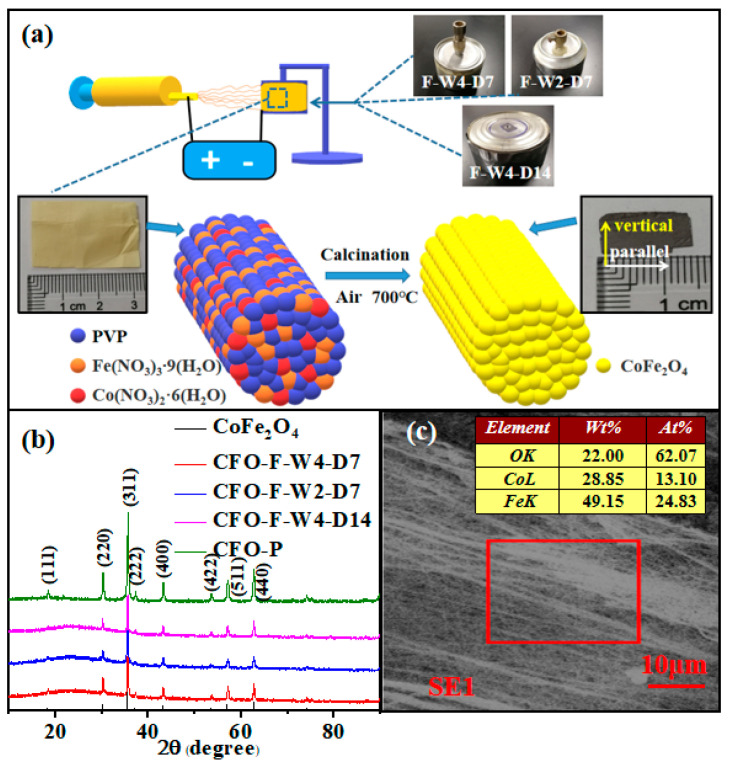
(**a**) A schematic diagram for the preparation of aligned nanofibers. (**b**) X-ray diffractometer (XRD) patterns of CoFe_2_O_4_ nanoparticles and nanofibers. (**c**) Low magnification scanning electron microscopy (SEM) photograph and element distribution of CoFe_2_O_4_ aligned nanofibers.

**Figure 2 materials-13-03860-f002:**
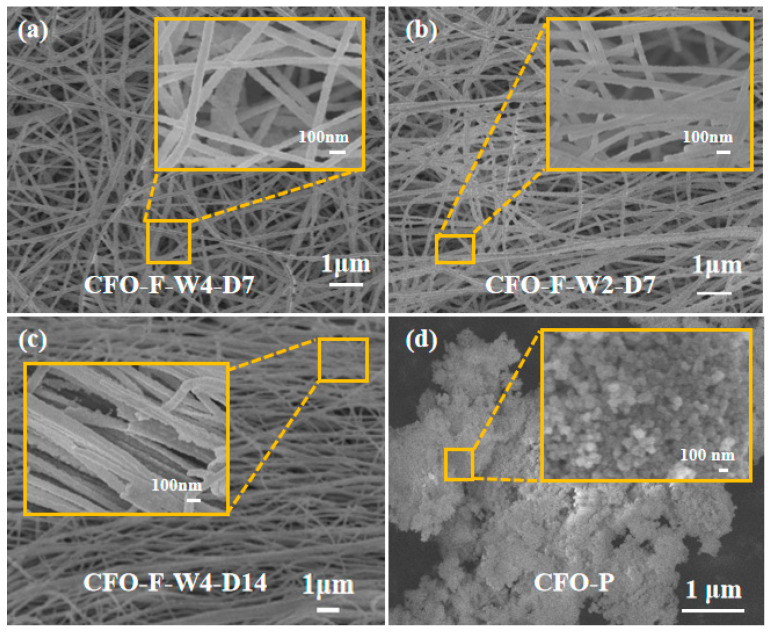
SEM micrographs of CoFe_2_O_4_ nanoparticles and nanofibers. (**a**) Microstructure of fiber CFO-F-W4-D7; (**b**) Microstructure of fiber CFO-F-W2-D7; (**c**) Microstructure of fiber CFO-F-W4-D14; (**d**) Morphology of CoFe_2_O_4_ particles.

**Figure 3 materials-13-03860-f003:**
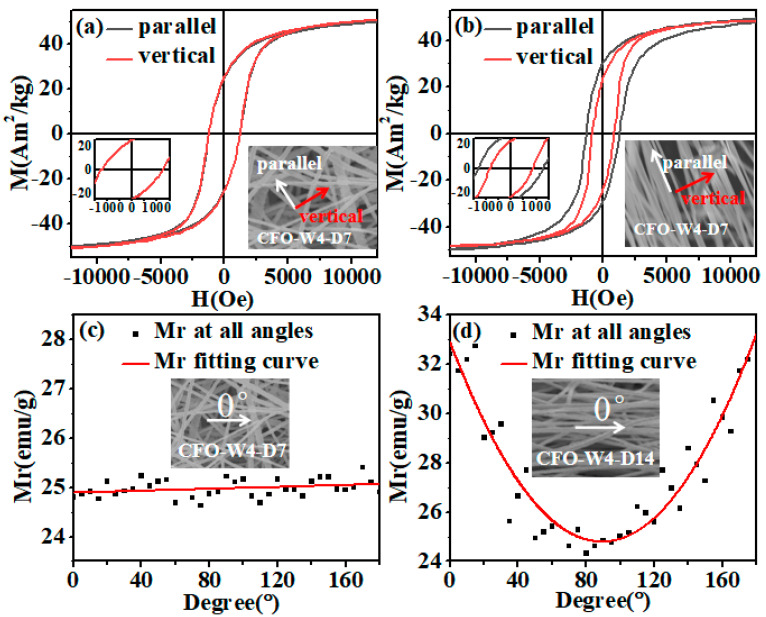
(**a**) Orthogonal axis hysteresis curves of non-oriented CoFe_2_O_4_ nanofibers (CFO-F-W4-D7) and (**b**) aligned nanofibers (CFO-F-W4-D14). In the lower right corner, a coercivity magnification image is inserted, and in the lower left corner, a SEM image showing the morphology of nanofibers is inserted. White arrows indicate that they are parallel to the axis, and red arrows indicate that they are perpendicular to the axis. It is the 0°~180° M_r_ in all directions of (**c**) non-oriented magnetic nanofibers (CFO-F-W4-D7) and (**d**) aligned magnetic nanofibers (CFO-F-W4-D14).

**Figure 4 materials-13-03860-f004:**
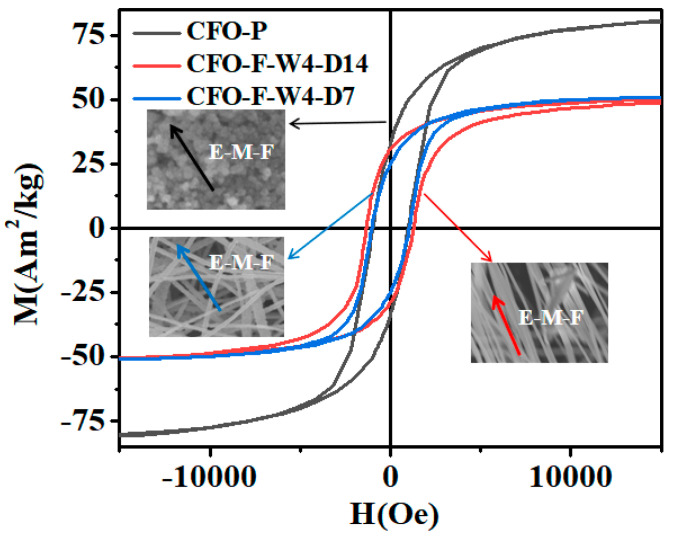
Loops of nanoparticles, aligned and non-oriented nanofibers. E-M-F is the direction of the external magnetic field.

**Figure 5 materials-13-03860-f005:**
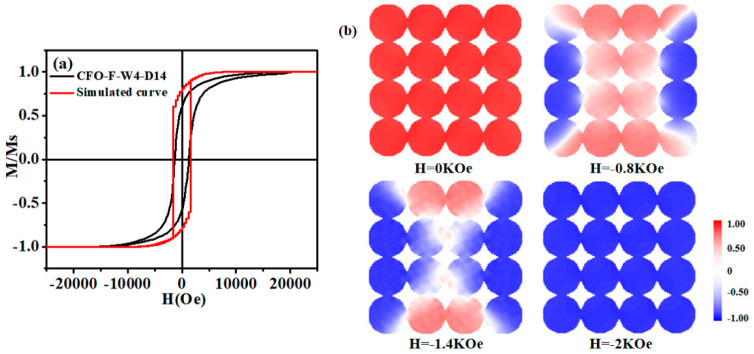
CoFe_2_O_4_ nanofibers array (**a**) hysteresis loop curve obtained by simulation and experiment. (**b**) Vertical view of the magnetic moment distribution in different states during demagnetization. The color scale represents the value of each component of the normalized magnetization.

**Table 1 materials-13-03860-t001:** Magnetic properties of aligned and non-oriented nanofibers.

Samples	M_s_ (Am^2^/kg)	H_c_ (Oe)	M_r_ (Am^2^/kg)	M_r_/M_s_
Non-oriented nanofibers (parallel-CFO-F-W4-D7)	49.95	1126.2	24.83	0.50
Non-oriented nanofibers (perpendicular-CFO-F-W4-D7)	50.67	1118.2	25.12	0.49
Aligned nanofibers (parallel-CFO-F-W4-D14)	50.89	1330.5	32.39	0.64
Aligned nanofibers (perpendicular-CFO-F-W4-D14)	51.11	857.2	24.80	0.48
CFO-P	80.23	979.3	32.33	0.40

**Table 2 materials-13-03860-t002:** Magnetic parameters of CoFe_2_O_4_ with different morphologies.

Morphology	Method	M_s_	H_c_	Reference
Spherical	Co-Precipitation Method	64.45 emu/g	681.04 Oe	[39]
Nanoparticles	Microwave Heating Method	23.88 emu/g	237 Oe	[40]
Nanowires	Electrodeposited	\	1300 Oe	[41]
Nanocables	Electrospinning	\	878 Oe	[42]
Nanowire Arrays	Anodic Aluminum Oxide Template	\	1100 Oe	[43]
Thin Films	Sol–Gel Method	200 emu/cm^3^	1000 Oe	[44]

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
