# Peer review of "Preparation and Magnetic Properties of CoFe_2_O_4_ Oriented Fiber Arrays by Electrospinning"

_materials, 2020, doi:10.3390/ma13173860_

Round 1

Reviewer 1 Report

The present submitted paper deals with the synthesis by electrospinning method of CoFe2O4 nanofibers and with their morphological and magnetic characterization.

I find that the scientific topic discussed in this paper is original and of scientific interest but I have some major criticisms about this article.

The main problem of this paper is that the numerous shortcomings in the English grammar and syntax heavily affect the understanding of the text. I ask the authors to check over all the manuscript.

Moreover, I find that the results description and discussion is not completely persuasive and I ask the authors to add further rigorous comments and considerations to the corresponding section.

In particular, the experimental data and results are presented in a too dispersive way and a more detailed, especially more connected and deeper discussion in the “Results and discussion” section is mandatory.

Author Response

Dear editors and reviewers:

Thank you for your letter and for the reviewers’ comments concerning our manuscript entitled “Preparation and magnetic properties of CoFe2O4 oriented fiber arrays by electrospinning

”ID(materials-878749). Those comments are all valuable and very helpful for revising and improving our paper, as well as the important guiding significance to our researches. We have studied comments carefully and have made correction which we hope meet with approval. Revised portion are marked in blue in the paper.

The main corrections in the paper and the reply to the reviewer’s comments are as following:

Responds to the reviewer’s comments:

Reviewer #1:

1.response to comment: The main problem of this paper is that the numerous shortcomings in the English grammar and syntax heavily affect the understanding of the text. I ask the authors to check over all the manuscript.

Response: We are very sorry for our negligence of that We've checked again to make the article more readable.

2.response to comment: Moreover, I find that the results description and discussion is not completely persuasive and I ask the authors to add further rigorous comments and considerations to the corresponding section.

In particular, the experimental data and results are presented in a too dispersive way and a more detailed, especially more connected and deeper discussion in the “Results and discussion” section is mandatory.

Response: We have made correction according to the reviewer’s comments.We have revised some contents and added more references to explain the phenomena found in the paper.

We tried our best to improve the manuscript and made some changes in the manuscript. These changes will not influence the content and framework of the paper. And here we did not list the changes but marked in blue in revised paper.

We appreciate for Editord/Reviewers’ warm work earnestly ,and hope that the correction will meet with approval.

Once again , thank you very much for your comments and suggestion.

Best regards

Reviewer 2 Report

Dear Editor,

The manuscript “Simple synthesis and magnetic characterization of CoFe2O4 nanofibers arrays” by Chen Cheng, Jianfeng Dai, Zengpeng Li and Wei Feng studies the fabrication of magnetic nanofibers based on electrospinning. Electrospinning allows to fabricate a mat of magnetic nanofibers. If no special steps are taken the mat consists of randomly oriented magnetic fibers. In many applications people wants to have oriented fibers. In the current paper authors tried to understand if one can get oriented fiber with collecting device representing a rotating drum. They found that linear velocity of the drum surface affects the orientation of the fiber in the mat. Generally, the faster the surface of the drum moves the more oriented the fibers become. Author prepare a few samples using different linear speed of the collecting drum and demonstrate using SEM images that higher speed leads to better alignment. Author also provide magnetic characterization of the samples and support it with modelling.

The dependence of the fiber alignment of the drum velocity is a well known effect (see for example review Composites Science and Technology 70 (2010) 703–718 and references there in:  Appl. Phys. Lett. 84, 1222 (2004), Nano Lett., Vol. 4, No. 11, 2004 and etc.). Authors of the present manuscript do not mention this and do not discuss what is new in their manuscript comparing to the works mentioned above or other works on the topic. So, from my point of view the novelty of the work is questionable and I cannot recommend it for publication.

Below, please find some additional comments on the paper.

  1. While authors do not mention this in introduction there are other ways to get aligned fiber. For example, by using collecting electrodes in the shape of two metal stripes as considered in Chem. Mater. 2007, 19, 3506-3511. Authors should discuss these other methods and compare their results with the results of other people.
  2. Magnetic field assisted fiber fabrication allows to create a very well aligned fibers (see for example, M.-C. Chen et al. / Acta Biomaterialia 9 (2013) 5562–5572). I think authors should compare their results to what was obtained by using magnetic field assisted fabrication. What is the benefit of the proposed method?
  3. There should be some references confusion. For example, Ref. 7 does not deal with microwave applications (as stated in the manuscript)
  4. Text (style and language) should be improved essentially. For example, in the very first sentence of the Abstract the combination “magnetic materials” is used three times. There are also many misprints, grammar errors and unclear sentences in the manuscript.
  5. In the introduction authors use the term Teylor cone. Can they explain what does it mean?
  6. Fig. 1(a) should be explained in more details. What are these corn-like structures? Why fibers consist of small spheres with different composition?
  7. In. Fig. 1(c) authors use some notations (such as OK, FeK, and etc.). These notations should be explained. Scale in Fig. 1(c) is not readable.

Author Response

Dear editors and reviewers:

Thank you for your letter and for the reviewers’ comments concerning our manuscript entitled “Preparation and magnetic properties of CoFe2O4 oriented fiber arrays by electrospinning

”ID(materials-878749). Those comments are all valuable and very helpful for revising and improving our paper, as well as the important guiding significance to our researches. We have studied comments carefully and have made correction which we hope meet with approval. Revised portion are marked in blue in the paper.

The main corrections in the paper and the reply to the reviewer’s comments are as following:

Responds to the reviewer’s comments:

Reviewer #2:

1.response to comment: While authors do not mention this in introduction there are other ways to get aligned fiber. For example, by using collecting electrodes in the shape of two metal stripes as considered in Chem. Mater. 2007, 19, 3506-3511. Authors should discuss these other methods and compare their results with the results of other people.

Response: We have rewritten this part according to the reviewer’s suggestion. We have added other articles that use electrospinning but different collection devices. The advantages and disadvantages of various methods are analyzed. The revised part has been marked in blue

2.response to comment: Magnetic field assisted fiber fabrication allows to create a very well aligned fibers (see for example, M.-C. Chen et al. / Acta Biomaterialia 9 (2013) 5562–5572). I think authors should compare their results to what was obtained by using magnetic field assisted fabrication. What is the benefit of the proposed method?

Response:We have rewritten this part according to the reviewer’s suggestion. The effect of external magnetic field on the collecting fiber is also discussed.

3.response to comment:There should be some references confusion. For example, Ref. 7 does not deal with microwave applications (as stated in the manuscript)

Response:We are very sorry for our incorrect writing. We re adjusted and checked the references.

4.response to comment:Text (style and language) should be improved essentially. For example, in the very first sentence of the Abstract the combination “magnetic materials” is used three times. There are also many misprints, grammar errors and unclear sentences in the manuscript.

Response: We are very sorry for our negligence of that We've checked again to make the article more readable

5.response to comment:In the introduction authors use the term Teylor cone. Can they explain what does it mean?

Response:As reviewer suggested that in order to better explain the term, some descriptions have been added in the paper.

6.response to comment:Fig. 1(a) should be explained in more details. What are these corn-like structures? Why fibers consist of small spheres with different composition?

Response:We think that the colored sphere in Figure 1 represents different components. Nanofibers were prepared under the constraint of polymer.

7.response to comment:In. Fig. 1(c) authors use some notations (such as OK, FeK, and etc.). These notations should be explained. Scale in Fig. 1(c) is not readable.

Response:K and L represent the line system of the element respectively. When the atomic number of the element to be analyzed is Z < 32, the K-line system is used. When the atomic number of the analyzed element is 32 < Z < 72, the L-line system is used. The M-line system is used when the atomic number of the element to be analyzed is Z > 72. In order to facilitate the understanding of the data, we have removed the identification of K and l, leaving only element symbols.

We tried our best to improve the manuscript and made some changes in the manuscript. These changes will not influence the content and framework of the paper. And here we did not list the changes but marked in blue in revised paper.

We appreciate for Editord/Reviewers’ warm work earnestly ,and hope that the correction will meet with approval.

Once again , thank you very much for your comments and suggestion.

Best regards

Reviewer 3 Report

Interesting paper with comprehensive wide study and promising future application. Authors posed an interesting problem in practical terms, successfully carried out many-sided experimental and theoretical research.

There are some mistakes in English

Line 126 the word “directionst” is written not correct.

line 148 “axes of nanofibers is perpendicular to and parallel to the long axes” fibers are plural! This sentence must be corrected.

Author Response

Dear editors and reviewers:

Thank you for your letter and for the reviewers’ comments concerning our manuscript entitled “Preparation and magnetic properties of CoFe2O4 oriented fiber arrays by electrospinning

”ID(materials-878749). Those comments are all valuable and very helpful for revising and improving our paper, as well as the important guiding significance to our researches. We have studied comments carefully and have made correction which we hope meet with approval. Revised portion are marked in blue in the paper.

The main corrections in the paper and the reply to the reviewer’s comments are as following:

Responds to the reviewer’s comments:

Reviewer #3:

1.response to comment:Line 126 the word “directionst” is written not correct.

Response: Special thanks to you for your good comments. We are very sorry for our incorrect writing.We have revised this and checked it.

2.response to comment:line 148 “axes of nanofibers is perpendicular to and parallel to the long axes” fibers are plural! This sentence must be corrected.

Response:We are very sorry for our negligence. We have revised this and checked it.

We tried our best to improve the manuscript and made some changes in the manuscript. These changes will not influence the content and framework of the paper. And here we did not list the changes but marked in blue in revised paper.

We appreciate for Editord/Reviewers’ warm work earnestly ,and hope that the correction will meet with approval.

Once again , thank you very much for your comments and suggestion.

Best regards

Reviewer 4 Report

In this work, the authors prepare cobalt ferrite nanofibers by electrospinning and they present and discuss the differences in their magnetic properties as a result of modifications in their synthetic parameters. They also compare their nanofibers with nanoparticles. This work has potential for publication, but some remarks have to be addressed first:

Manuscript title: It is too simple. ‘Simple synthesis and magnetic characterization’… Make it more appealing, write about electrospinning, about oriented fibers etc, for example.

Abstract, line number 15: Which ‘were found to have similar diameters’? The different fiber samples? The different fibers among a single sample? You need to be more specific.

Line 19: Again, be specific: The shape anisotropy and interaction between NPs how does they affect the magnetization and coercivity values? In positive way? In negative way? And with which mechanism?

Line 22: After mentioning magnetic parameter values and simulations, you write that this preparation method is low-cost. Do you think that this is for the last phrase of your Abstract or maybe you have to put it somewhere above?

Page 2, line 48: You can cite also more papers when you present the advantages of electrospinning technology to produce magnetic fibers. For example, there is a recent paper on electrospun, oriented Ni-Fe nanofibers: Frontiers in Chemistry, year 2020, volume 8, article number 47.

Page 2, line 61: ‘Similar’ or ‘homogeneous’?

Line 64: Define OOMMF.

Line 66: Please write also the purities of the products.

Page 2, line 73: The heat treatment was done in air or under inert atmosphere? The same question applies for what is written at the line 82.

Page 4, line 104: Change ‘diameters’ to ‘mean diameters’. Also, one decimal digits is enough. For example write ’94.6’ instead of ’94.63’ etc.

Line 107: What do you mean by ‘higher directionality’?

Table 1: The Mr/Ms ratio can have up to two decimal digits. 4 decimals are too many.

Page 6, line 139: What do you mean that the sample D7 is isotropic? It may be non-oriented, and this can influence its magnetic properties, but how can it be ‘isotropic’? Its morphology is anisotropic, right?

Line 166: The phrase starting with ‘In Ms’ is not clear. Please rephrase.

Line 197: Apart from comparing your experimental value of coercivity with simulations, you can compare it also with more CoFe2O4 nanofibers and other cobalt ferrite nanostructures from the literature, prepared by electrospinning or other methods.

You have previously published works on CoFe2O4@NiO and NiFe2O4@CoFe2O4 nanofibers. Can you describe the similarities and the differences of the current work in respect to those ones? Especially in the synthetic part. OK, here you do not use DMF. Please describe also any rest differences.

References 12 and 17 are the same. Therefore please delete the reference 17 and re-arrange the numbering of references in both main text and list of references

Author Response

Dear editors and reviewers:

Thank you for your letter and for the reviewers’ comments concerning our manuscript entitled “Preparation and magnetic properties of CoFe2O4 oriented fiber arrays by electrospinning

”ID(materials-878749). Those comments are all valuable and very helpful for revising and improving our paper, as well as the important guiding significance to our researches. We have studied comments carefully and have made correction which we hope meet with approval. Revised portion are marked in blue in the paper.

The main corrections in the paper and the reply to the reviewer’s comments are as following:

Responds to the reviewer’s comments:

Reviewer #4:

1.response to comment:Manuscript title: It is too simple. ‘Simple synthesis and magnetic characterization’… Make it more appealing, write about electrospinning, about oriented fibers etc, for example.

Response: It is really true as reviewer suggested that It is too simple. In order to make the topic more attractive, we rewrote the title as Preparation and magnetic properties of CoFe2O4 oriented fiber arrays by electrospinning

2.response to comment:Abstract, line number 15: Which ‘were found to have similar diameters’? The different fiber samples? The different fibers among a single sample? You need to be more specific.

Response:We have revised the original sentence. “After the same annealing at 700 ℃, it is found that the diameter of fibers collected by different collecting drums is similar”

3.response to comment:Line 19: Again, be specific: The shape anisotropy and interaction between NPs how does they affect the magnetization and coercivity values? In positive way? In negative way? And with which mechanism?

Response:We have revised the original sentence. “Because of the anisotropy of the shape and the interaction between the particles, the Hc can not be enhanced. Therefore, the Ms and Hc of the nanoparticles are 80.23 Am2/kg and 979.3 Oe, respectively.” So here it plays a negative role.

4.response to comment:Line 22: After mentioning magnetic parameter values and simulations, you write that this preparation method is low-cost. Do you think that this is for the last phrase of your Abstract or maybe you have to put it somewhere above?

Response: We have revised and deleted them according to the reviewers' opinions.

5.response to comment:Page 2, line 48: You can cite also more papers when you present the advantages of electrospinning technology to produce magnetic fibers. For example, there is a recent paper on electrospun, oriented Ni-Fe nanofibers: Frontiers in Chemistry, year 2020, volume 8, article number 47.

Response:In order to better introduce the advantages of electrospinning technology in the production of magnetic fibers, we added new references. The advantages and disadvantages of using electrospinning technology and different fiber directional collection devices are discussed.

6.response to comment:Page 2, line 61: ‘Similar’ or ‘homogeneous’?

Response: This means that the diameter of the fiber is similar to that of the nanoparticles. Of course, fibers and particles are also CoFe2O4 materials.

7.response to comment:Line 64: Define OOMMF.

Response: It has been modified in this paper. OOMMF stands for Object Oriented Micro Magnetic Framework .

8.response to comment:Line 66: Please write also the purities of the products.

Response: In this paper, all other chemicals used in this work were of analytical grade.

9.response to comment:Page 2, line 73: The heat treatment was done in air or under inert atmosphere? The same question applies for what is written at the line 82.

Response: In this paper, the treatment was carried out in the air atmosphere.

10.response to comment:Page 4, line 104: Change ‘diameters’ to ‘mean diameters’. Also, one decimal digits is enough. For example write ’94.6’ instead of ’94.63’ etc.

Response: We have made correction according to the reviewer’s comments.

11.response to comment:Line 107: What do you mean by ‘higher directionality’?

Response: That means CFO-W2-D7 has higher degree of orientation than CFO-W4-D7.

12.response to comment:Table 1: The Mr/Ms ratio can have up to two decimal digits. 4 decimals are too many.

Response: We have made correction according to the reviewer’s comments.

13.response to comment:Page 6, line 139: What do you mean that the sample D7 is isotropic? It may be non-oriented, and this can influence its magnetic properties, but how can it be ‘isotropic’? Its morphology is anisotropic, right?

Response: We have rewritten this part according to the reviewer’s suggestion.

14.response to comment:Line 166: The phrase starting with ‘In Ms’ is not clear. Please rephrase.

Response: We are very sorry for our incorrect writing

15.response to comment:Line 197: Apart from comparing your experimental value of coercivity with simulations, you can compare it also with more CoFe2O4 nanofibers and other cobalt ferrite nanostructures from the literature, prepared by electrospinning or other methods.

Response: In order to better explain the phenomena in this paper, we added more literatures according to the opinions of commentators to compare the effects of different morphologies on magnetic nanomaterials.

16.response to comment:You have previously published works on CoFe2O4@NiO and NiFe2O4@CoFe2O4 nanofibers. Can you describe the similarities and the differences of the current work in respect to those ones? Especially in the synthetic part. OK, here you do not use DMF. Please describe also any rest differences.

Response: Because of the core-shell structure published before. In order to complete the preparation in one step, the selection of precursor solvent for core and shell materials needs to be more precise. It is better to use polar and nonpolar liquids to make them incompatible and to obtain core-shell nanomaterials more conveniently. However, this paper prepared fiber materials, in order to better production and environmental friendliness, so only choose alcohol and water as solvent.

17.response to comment:References 12 and 17 are the same. Therefore please delete the reference 17 and re-arrange the numbering of references in both main text and list of references

Response: We are very sorry for our incorrect writing. We have revised and edited the references

We tried our best to improve the manuscript and made some changes in the manuscript. These changes will not influence the content and framework of the paper. And here we did not list the changes but marked in blue in revised paper.

We appreciate for Editord/Reviewers’ warm work earnestly ,and hope that the correction will meet with approval.

Once again , thank you very much for your comments and suggestion.

Best regards

Round 2

Reviewer 1 Report

I find the authors have sufficiently fulfilled all the requirements I pointed out in my previous report, therefore I recommend the publication of the present paper in Materials.

Author Response

Special thanks to you for your good comments.

Reviewer 2 Report

Dear Editor,

I reviewed the revised manuscript “Simple synthesis and magnetic characterization of CoFe2O4 nanofibers arrays” by Chen Cheng, Jianfeng Dai, Zengpeng Li and Wei Feng. Authors introduced some changes into the manuscripts. However, the novelty of the results and the place of this work among other are unclear from the manuscript. The method of spinning using the rotating drum is well known. Magnetic properties of nanofibers produced by different methods including electrospinning are well studied as well (see for example Journal of Materials Science volume 51, pages 885–892(2016) and reference therein). The statement that the coercive field of the nanofiber is higher than that of the spherical nanoparticle is obvious. Author should discuss what is known about the magnetic properties of the nanofibers in the introduction and explain what new they found in their research. Therefore, I can not recommend the manuscript in the present form for the publication.

Still the manuscript is prepared with a little care about readers.

  • 38. What is Journal of Physics D Applied Physics A Europhysics Jpurnal? Is this a journal title?
  • 27. What is “Journal of Applied Polymerence” ?
  • The language is still need to be improved.
  1. The first sentence of the introduction: “ferrite … are known”
  2. Second paragraph of the introduction: What does term “magnetic difference” mean?
  3. “Method of collecting … by electrode method…”
  4. ….
  • Paragraph 5 of the introduction. Periods appears in the middle of the sentences.
  • Scale in Fig. 1 is still not readable. Notation ZAF is not explained.
  • How is the last sentence of the conclusion related to all previous sentences?
  • And etc.

Author Response

Dear editors and reviewers:

Thank you for your letter and for the reviewers’ comments concerning our manuscript entitled “Preparation and magnetic properties of CoFe2O4 oriented fiber arrays by electrospinning”ID (materials-878749). those comments are all valuable and very helpful for revising and improving our paper, as well as the important guiding significance to our researches. We have studied comments carefully and have made correction which we hope meet with approval. Revised portion are marked in blue in the paper.

The main corrections in the paper and the reply to the reviewer’s comments are as following:

1.Responds to the reviewer’s comments: 38. What is Journal of Physics D Applied Physics A Europhysics Jpurnal? Is this a journal title?27. What is “Journal of Applied Polymerence” ?

Reviewer :We are very sorry that we didn't write it correctly. It has been modified according to the author's specification.

2.Responds to the reviewer’s comments: The language is still need to be improved.The first sentence of the introduction: “ferrite … are known”. Second paragraph of the introduction: What does term “magnetic difference” mean?. “Method of collecting … by electrode method…”

Reviewer :We have made correction according to the reviewer’s comments. We have made a detailed revision of the article, and made corresponding improvements in grammar and word use.

3.Responds to the reviewer’s comments:Paragraph 5 of the introduction. Periods appears in the middle of the sentences. Scale in Fig. 1 is still not readable. Notation ZAF is not explained.

Reviewer :We have rewritten this part according to the reviewer’s suggestion. ZAF is a modified method for quantitative analysis with EDS. ZAF correction method is the first letter of the three coefficients (Z: atomic number A: absorption coefficient F: fluorescence coefficient). This section has been deleted for a clearer presentation of the results.

4.Responds to the reviewer’s comments: How is the last sentence of the conclusion related to all previous sentences?

Reviewer :We have made correction according to the reviewer’s comments.

We tried our best to improve the manuscript and made some changes in the manuscript. These changes will not influence the content and framework of the paper. We appreciate for Editord/Reviewers’ warm work earnestly ,and hope that the correction will meet with approval.

Once again , thank you very much for your comments and suggestion.

Best regards,